



# The Role of Eddy-Eddy Interactions in Sudden Stratospheric Warming Formation

Erik Anders Lindgren[1] and Aditi Sheshadri[1]

[1]Department of Earth System Science, Stanford University, Stanford, CA, USA.

**Correspondence:** Erik A. Lindgren (ealindgr@stanford.edu)

**Abstract.** The effects of eddy-eddy interactions on sudden stratospheric warming formation are investigated using an idealized atmospheric general circulation model, in which tropospheric heating perturbations of zonal wave numbers 1 and 2 are used to produce planetary scale wave activity. Eddy-eddy interactions are removed at different vertical extents of the atmosphere in order to examine the sensitivity of stratospheric circulation to local changes in eddy-eddy interactions. We show that the effects

of eddy-eddy interactions on sudden warming formation, including sudden warming frequencies, are strongly dependent on the wave number of the tropospheric forcing and the vertical levels where eddy-eddy interactions are removed. Significant changes in sudden warming frequencies are evident when eddy-eddy interactions are removed even when the lower stratospheric wave forcing does not change, highlighting the fact that the upper stratosphere is not a passive recipient of wave forcing from below. We find that while eddy-eddy interactions are required in the troposphere and lower stratosphere to produce displacements

when wave number 2 heating is used, both splits and displacements can be produced without eddy-eddy interactions in the troposphere and lower stratosphere when the model is forced by wave number 1 heating. We suggest that the relative strengths of wave numbers 1 and 2 vertical wave flux entering the stratosphere largely determine the split and displacement ratios when wave number 2 forcing is used, but not wave number 1.

## 1 Introduction

Sudden stratospheric warmings (SSWs) are dynamical events that can occur during hemispheric winter and which result in a collapse of the stratospheric polar vortex. During SSWs the temperature of the middle and upper polar stratosphere increases by over 30 K over a period of days (Butler et al., 2015), and the strongest SSWs, often called major SSWs, are usually defined by a reversal of the zonal mean westerlies at 60° and 10 hPa (Charlton and Polvani, 2007). Accurate simulation of SSWs in models is crucial to capture the variability of surface climate in midlatitudes, since the effects of SSWs migrate down to

the troposphere and can impact surface weather up to two months after onset (e.g., Baldwin and Dunkerton, 2001). During SSWs the stratospheric polar vortex is either displaced from the pole as a single entity or split into two daughter vortices. These two types of SSWs are known as displacements and splits, and they are dominated by zonal wave number 1 and 2 disturbances, respectively. It has been suggested that splits and displacements are dynamically distinct (Charlton and Polvani, 2007; Matthewman et al., 2009), and some studies claim that they may have different surface impacts (e.g., Mitchell et al.,

2013; Seviour et al., 2013; Lindgren et al., 2018), while others have found that differences in surface impacts are not consistent





for different split and displacement classifications, and that large numbers of events are required to distinguish any differences (Maycock and Hitchcock, 2015). Major SSWs occur about every other year in the Northern Hemisphere and are roughly equally distributed between splits and displacements (Charlton and Polvani, 2007; Lindgren et al., 2018, and others). Only one major SSW has been observed in the Southern Hemisphere (e.g., Allen et al., 2003).

Despite their importance for Northern Hemisphere climate, the dynamics behind SSW generation remains poorly understood. SSWs can occur when waves propagate from the troposphere to stratosphere where they break and deposit momentum (e.g., Matsuno, 1971), and from this perspective SSWs can be considered wave-mean flow interactions. It has long been known that SSW-like zonal mean wind variations can occur in model setups as simple as one-dimensional $\beta$-plane models (e.g., Holton and Mass, 1976; Yoden, 1990). SSW generation in general circulation models (GCMs) and the observed atmosphere,

however, is likely much more complicated. Although anomalously strong tropospheric wave forcing can produce SSWs, SSW-like events have been found to occur in idealized GCMs with suppressed tropospheric variability (Scott and Polvani, 2004, 2006), and recent research has shown that the majority of SSWs in both reanalysis data (Birner and Albers, 2017) and one idealized GCM (Lindgren et al., 2018) occur without anomalous tropospheric wave forcing. Birner and Albers (2017) argued that the dynamical processes responsible for most SSWs occur just above the tropopause, even though the wave forcing responsible

for the events originate near the surface. The importance of stratospheric processes in SSW dynamics was also emphasized by Hitchcock and Haynes (2016), who found that the evolution of the stratospheric mean state during SSWs is crucial in determining the wave flux during the warming. Given the complexity of SSW generation it is not possible to predict how frequently, or even if, SSWs will occur in a given model setup *a priori*. GCMs are therefore often extensively tuned to produce realistic SSW frequencies.

Exactly which dynamical processes are responsible for vortex splits and displacements is also unclear. Since displacements and splits are zonal wave number 1 and 2 disturbances, respectively, one could expect that the zonal wave number of the wave flux propagating from the troposphere will determine the type of SSW produced. Large-scale topography of a single, zonal wave number has often been used to produce Northern Hemisphere winter-like stratospheric variability in idealized GCMs, and in such cases the wave number of the topography does indeed strongly influence the type of SSW produced, with wave number

1 (wave 1) topography favoring displacements (Martineau et al., 2018) and wave number 2 (wave 2) topography producing mostly or only splits (Gerber and Polvani, 2009; Sheshadri et al., 2015; Martineau et al., 2018; Lindgren et al., 2018). Recently, however, Lindgren et al. (2018) showed that splits and displacements occur in comparable amounts when an idealized GCM is forced with wave 1 or wave 2 tropospheric heating perturbations. Since the tropospheric forcings are of pure wave 1 or 2 format these results indicate that some wave-wave (eddy-eddy) interaction could be taking place somewhere between the waves being

forced the troposphere and the waves breaking in the stratosphere.

Idealized models are useful tools when investigating specific dynamical processes, and simple models have been frequently used in previous studies of SSW dynamics. After one-dimensional $\beta$-plane models (e.g., Holton and Mass, 1976; Yoden, 1990) the next step in the model hierarchy includes the effects of eddy-eddy interactions (EEIs), and research about the role of EEIs in stratospheric dynamics dates back to the 1980s. Lordi et al. (1980) applied wave 1 and 2 geopotential height forcings at the

lower boundary of a primitive equation spectral model of the stratosphere to simulate SSWs. They used spectral truncation to





remove interactions between wavenumbers, and only allowed the wave number of the forcing to interact with the main flow. They concluded that nonlinear interactions are important for the evolution of the mean flow and temperature fields in the middle and upper stratosphere, especially when wave 1 forcing was used. Austin and Palmer (1984) used a primitive equation model of the stratosphere and mesosphere to investigate the role of nonlinear effects in setting up the monthly mean wave amplitudes

in the stratosphere during December 1980, and concluded that EEIs cannot be ignored for accurate simulations of the middle atmosphere.

O'Neill and Pope (1988) carried out an extensive study of linear and nonlinear disturbances in the stratosphere resulting from tropospheric perturbations, with the same primitive equations model that Austin and Palmer (1984) used. They found that linear theories of wave propagation and wave-mean flow interaction were useful when the imposed lower boundary forcing was

weak, but that the stratospheric flow was highly nonlinear in the latter stages of simulations with strong forcing. Their results suggested that major stratospheric warmings evolve from highly asymmetric states through complicated nonlinear interactions, rather than simple wave-mean flow interactions.

Robinson (1988) simulated a minor wave 1 stratospheric warming in a primitive equation model, while running the model in a fully nonlinear mode as well as in a quasi-linear mode, with waves of only one wave number and the zonal flow. The model

runs were 60 days long following the spin-up period. He found that irreversible modifications of the potential vorticity field were stronger and more localized when EEIs were allowed, and that the differences between the nonlinear and quasi-linear model runs come from modifications of the interactions between wave 1 and the zonal flow by shorter waves. The interactions between shorter waves and the mean flow were found to be of much less importance when accounting for the differences between the experiments.

There is also observational evidence to suggest that EEIs are largely responsible for modulating the relative strengths of wave 1 and wave 2 disturbances in the stratosphere. Smith et al. (1983) used satellite data to investigate the importance of wave-mean flow interactions versus EEIs in the Northern Hemisphere during the 1978-79 winter, and found that vacillations between wave 1 and wave 2 in the geopotential height field could be largely attributed to EEIs in the stratosphere, rather than forcing from the tropopause region.

The authors mentioned above have shown that EEIs play an important role in stratospheric dynamics in general and SSW dynamics in particular, but many questions regarding the role of EEIs in SSW dynamics remain unanswered. One major restriction of the previous studies is that spectral truncation was used to eliminate EEIs, while keeping only the interactions between a single wave number and the mean flow (Lordi et al., 1980; Robinson, 1988). The climatological wave flux in the observed stratosphere contains both wave 1 and wave 2 components (e.g., Birner and Albers, 2017), both of which can interact

with the mean flow. Given that displacements and splits are wave 1 and wave 2 disturbances, respectively, removal of either of the major stratospheric wave numbers along with EEIs will enable only one type of SSW to form, even though SSWs in the observed atmosphere are roughly equally distributed between splits and displacements (Charlton and Polvani, 2007; Lindgren et al., 2018, and others). Another limitation of previous work is the short temporal extent of the model runs, due to the limited computing resources of the 1980s. It is known that EEIs are important both for SSW generation (O'Neill and Pope, 1988) and





the evolution of the stratospheric mean state during SSWs (Hitchcock and Haynes, 2016) but the effect that EEIs have on SSW frequencies has not been investigated, and it is not clear *a priori* whether EEIs act to enhance or diminish SSW generation.

Furthermore, previous work has removed EEIs throughout the vertical extent of the models. Birner and Albers (2017) emphasized the importance of dynamics in the 300-200 hPa region or just above the tropopause for SSW formation. Polvani and Waugh (2004) found that anomalous wave fluxes at 100 hPa and further down in the troposphere precede SSWs and

deduced that the origin of SSWs can be found in the troposphere (although they acknowledged the fact that the stratosphere may play a role in modulating the events). These results indicate that dynamical processes in the troposphere and lower stratosphere are crucial in SSW formation, which raises the question as to whether or not the role of EEIs in SSW dynamics varies accordingly. The recent results of (Lindgren et al., 2018) show that both splits and displacements can form with a tropospheric forcing of a single wave number, indicating that EEIs could act to transfer energy from the wave number of the

forcing to the other major stratospheric wave number somewhere between the troposphere and the stratosphere. In order to understand the importance of EEIs for SSW generation as well as split and displacement distributions at different vertical levels in the atmosphere, a different method of removing EEIs must be used.

More recently, the effects of EEIs on atmospheric dynamics has been investigated in idealized models by calculating the advection of eddy fluctuations by eddy winds, and replacing them with their zonal mean values. Unlike the spectral truncation

mentioned in the papers above this method allows climatological wave flux of all wave numbers to interact with the mean flow, and it retains much of the zonal mean climatology of (nonlinear) control runs. The method has been successfully used to investigate tropospheric dynamics by several authors: O'Gorman and Schneider (2007) removed EEIs an idealized GCM and found that the atmospheric kinetic energy spectrum retained its wave number dependence when EEIs were removed, even though this dependence was previously thought to be determined by EEIs. Chemke and Kaspi (2016) showed that EEIs in an

idealized GCM actually decrease the number of eddy-driven jets in the atmosphere by narrowing the latitudinal region where zonal jets can appear. This method has also proven useful when applied to advanced theory of jet dynamics in beta-plane models (e.g., Srinivasan and Young, 2012; Tobias and Marston, 2013; Constantinou et al., 2014).

In this paper we investigate the role of EEIs in SSW formation by removing the effects of zonal EEIs in an idealized GCM, using the method of O'Gorman and Schneider (2007). We use the model output produced with heating wave 1 (H1) and wave

2 (H2) tropospheric forcing from Lindgren et al. (2018), both of which have been shown to produce splits and displacements in comparable amounts. We perform model runs under three additional settings for each forcing: one without EEIs anywhere, one without EEIs in the troposphere and lower stratosphere, and one without EEIs in the middle and upper stratosphere. The latter two, hereafter referred to as the mixed runs, allow us to investigate the effects of removing EEIs above (below) the pressure levels that Polvani and Waugh (2004) and Birner and Albers (2017) highlighted as crucial for SSW generation, while

still allowing EEIs below (above). By comparing the results of the mixed runs to the fully nonlinear control runs and the model runs without EEIs anywhere, we can investigate the importance of EEIs in the middle and upper stratosphere when the climatology (and hence mean wave forcing) in the troposphere and lower stratosphere remains unchanged. We use the model runs to answer three questions related to EEIs and SSW formation that have not been investigated before:

1. To what extent do EEIs affect SSW frequencies?





2. To what extent are the numbers of splits and displacements, and their ratios, affected by EEIs?

3. How do EEIs affect the numbers of SSWs, splits and displacements when the lower stratospheric wave forcing does not change?

We find that the effects of EEIs on SSWs are strongly dependent on the wave number of the tropospheric forcing. We show that removing EEIs can significantly alter SSW frequencies, and that whether EEIs increase or decrease SSW frequencies depends on the tropospheric forcing that is used and the vertical levels where EEIs are removed. Significant changes in SSW frequencies between model runs with and without EEIs in the middle and upper stratosphere are obtained even though the lower stratospheric wave forcings of the model runs do not change, highlighting the fact that the stratosphere is not a passive recipient of lower stratospheric wave forcing. We further find that while EEIs are required in the troposphere and lower stratosphere to produce displacements when wave 2 forcing is used, both splits and displacements can be produced without EEIs in the troposphere and lower stratosphere when the model is forced by wave 1 heating.

Section 2 describes the method used to remove the effects of EEIs, and the way it was implemented in the model runs. Section 3 describes changes in climatology caused by removal of EEIs. Section 4 compares the SSW frequencies, split and displacement ratios, and polar vortex strength variabilities of the eight different model runs. A discussion of the results and conclusions can be found in Sect. 5.

## 2 Removal of eddy-eddy interactions

Following O'Gorman and Schneider (2007), we calculated the tendency due to EEIs (the *eddy-eddy tendency*), subtracted it from the total tendency of horizontal wind and temperature, and added the zonal mean value of the eddy-eddy tendency (the *mean tendency*) to the total tendency equation. This method removes interactions between zonal waves. This substitution was described by O'Gorman and Schneider (2007) by using the equation for temperature tendency as an example. In the control run the evolution is

$$\begin{aligned}
\frac{\partial T}{\partial t} &= -v\frac{\partial T}{\partial y} + ..., \\
&= -\bar{v}\frac{\partial \overline{T}}{\partial y} - \bar{v}\frac{\partial T'}{\partial y} - v'\frac{\partial \overline{T}}{\partial y} - v'\frac{\partial T'}{\partial y} + ...,
\end{aligned} \tag{1}$$

where overbars denote zonal means while primes show deviations from the zonal mean (eddies). Only the terms related to meridional advection of temperature have been written out. The last term in Eq. (1) describes the advection of temperature eddies due to meridional wind eddies. In the model runs where EEIs are not allowed Eq. (1) becomes

$$\begin{aligned}
\frac{\partial T}{\partial t} &= -v\frac{\partial T}{\partial y} + ..., \\
&= -\bar{v}\frac{\partial \overline{T}}{\partial y} - \bar{v}\frac{\partial T'}{\partial y} - v'\frac{\partial \overline{T}}{\partial y} - \overline{v'\frac{\partial T'}{\partial y}} + ...,
\end{aligned} \tag{2}$$





where the contribution due to EEIs has been replaced by its zonal mean value.

Two model runs from Lindgren et al. (2018) are used in this paper. The GCM is a dry, hydrostatic, global primitive equation model with T42 resolution in the horizontal and 40 vertical $\sigma$ levels, where $\sigma = p/p_s$. There are no convection or radiation schemes in the model, and temperature is relaxed towards a zonally symmetric temperature profile through Newtonian relaxation. The temperature profile is symmetric about the equator in the troposphere, but set to perpetual Northern Hemisphere midwinter conditions in the stratosphere. The transition between tropospheric and stratospheric equilibrium temperature profiles occurs at 200 hPa. Tropospheric diabatic heating perturbations are used to produce Northern Hemisphere winter-like stratospheric variability. The reader is referred to Lindgren et al. (2018) for more information about the model and heating perturbations. In addition to the H1 and H2 runs from Lindgren et al. (2018) another three runs were performed for each forcing wave number, where each of the three additional runs removed EEIs in different parts of the atmosphere. All model set ups were run under Northern Hemisphere winter conditions for 31,100 days, and the last 30,000 days were used for the analysis. The vertical structures of where EEIs are permitted and removed can be found in Figs. 1 and 2.

In the control runs (H1 and H2; black line in Figs. 1 and 2) EEIs are allowed everywhere. In the no EEIs-anywhere runs (NE1 or NE2 depending on the wave number of the forcing; red line) the effects of EEIs were removed at each pressure level. In the mixed runs the model is set up to switch between allowing and not allowing EEIs linearly with pressure. The transition occurs between 50 and 30 hPa. In the case of the runs with no EEIs allowed in the troposphere and lower stratosphere (hereafter shortened NEt1 or NEt2; green line) the following substitution is made:

$$
\frac{\partial T}{\partial t} = \begin{cases} \dfrac{\partial T}{\partial t} - \overline{v'\dfrac{\partial T'}{\partial y}} + v'\dfrac{\partial T'}{\partial y}, \ p > p_1, \\[2ex] \dfrac{\partial T}{\partial t} + \left(1 - \dfrac{p_1-p}{p_1-p_2}\right) \cdot \left(-\overline{v'\dfrac{\partial T'}{\partial y}} + v'\dfrac{\partial T'}{\partial y}\right), \ p_2 \leq p \leq p_1, \\[2ex] \dfrac{\partial T}{\partial t}, \ p < p_2. \end{cases} \tag{3}
$$

In the above equation the temperature tendency from Eqs. 1 and 2 has been used as an example. $p_1 = 50$ hPa and $p_2 = 30$ hPa. Similarly, when EEIs are not allowed in the upper stratosphere (hereafter shortened NEs1 or NEs2; blue line) the equation describing the substitution is

$$
\frac{\partial T}{\partial t} = \begin{cases} \dfrac{\partial T}{\partial t}, \ p > p_1, \\[2ex] \dfrac{\partial T}{\partial t} + \dfrac{p_1-p}{p_1-p_2} \cdot \left(-\overline{v'\dfrac{\partial T'}{\partial y}} + v'\dfrac{\partial T'}{\partial y}\right), \ p_2 \leq p \leq p_1, \\[2ex] \dfrac{\partial T}{\partial t} - \overline{v'\dfrac{\partial T'}{\partial y}} + v'\dfrac{\partial T'}{\partial y}, \ p < p_2. \end{cases} \tag{4}
$$

The mixed runs enable an investigation of the effects of EEIs in different regions of the atmosphere. Although it may seem an obvious choice to put the transition region around the tropopause this alters the climatologies of the mixed runs significantly compared to the control runs, something that is likely caused by significant changes in the strong climatological





wave convergence in the tropopause region (see Figs. S2 and S3 in the supporting information). Instead, the 50 hPa and 30 hPa levels were chosen as start and end points of the transition region. This choice has two strong advantages over the tropopause region: first, it is an unusually calm region of the atmosphere in terms of wave activity and changes in wave interactions do not

affect the climatology as strongly. Second, the pressure levels where lower stratospheric wave forcing has been highlighted as important for SSWs range from 300-200 hPa (Birner and Albers, 2017) to 100 hPa (Polvani and Waugh, 2004). The choice of a transition region between 50 and 30 hPa as opposed to the tropopause means that we can keep what is thought to be the most important wave forcing identical between model runs that allow (remove) EEIs everywhere and turn EEIs off (on) in the middle and upper stratosphere.

## 3   Climatology in the absence of EEIs

Figure 2 shows the climatological zonal mean zonal wind for the four model runs with wave 2 heating, along with panels indicating the pressure levels where EEIs are allowed. Zonal mean zonal winds for model runs with wave 1 heating can be found in the supporting information (Fig. S1). A comparison of H2 (Fig. 2a) to NE2 (Fig. 2b) shows that the model retains much of the climatological zonal mean zonal wind structure even in the absence of EEIs, although with some notable exceptions.

For one, the polar night jet is much more separated from the tropospheric jet when EEIs are not allowed in the troposphere and lower stratosphere (panels b and d). The jet strength, however, is largely unaffected in NE2. This is not the case in all model runs, especially NEs2 (Fig. 2c) compared to the other model runs with wave 2 heating. The area where the zonal mean zonal wind approaches zero $\mathrm{ms}^{-1}$ found in the equatorial lower stratosphere in the control runs is not reproduced when EEIs are not allowed in the troposphere and lower stratosphere (panels b and d), indicating that EEIs play an important role in this area.

Furthermore, there are two tropospheric jets in the Southern Hemispheres of these runs. O'Gorman and Schneider (2007) and Chemke and Kaspi (2016) also obtained additional tropospheric jets in their models when removing EEIs, and Chemke and Kaspi (2016) showed that EEIs decrease the number of eddy-driven jets in the atmosphere.

Comparisons between the mixed runs and H2 or NE2 show that changes in the middle and upper stratosphere have very little influence on the climatology of the troposphere and lower stratosphere. When EEIs are allowed in the troposphere and lower

stratosphere only (NEs2; Fig. 2c) the zonal mean zonal winds at pressure levels below 50 hPa are very similar to those of the control run (H2; Figs. 2a). Similarly, when EEIs are not allowed in the troposphere and lower stratosphere (NEt2; Fig. 2d) the climatology at pressure levels below 50 hPa is very similar to that of the model run where EEIs are not allowed anywhere (NE2; Fig. 2b). The same is true when wave 1 heating is used; see Fig. S1.

To investigate the changes in wave activity caused by removing EEIs we calculated the climatological wave 1 and wave

2 components of vertical Eliassen-Palm (EP) flux ($F_p$) and divergence of EP flux for the eight model runs. The calculations were based on Edmon et al. (1980), and are identical to those found in Lindgren et al. (2018). Figures S2 through S5 in the supporting information show the wave 1 and wave 2 components of the two quantities. The most important result of the EP flux figures is that, just like the zonal mean zonal wind in Fig. 2, the divergence of EP flux and $F_p$ at pressure levels greater than 50 hPa depend on whether or not EEIs are allowed at these levels: NE1 and NE2 look very similar to NEt1 and NEt2,



while H1 and H2 look like NEs1 and NEs2 in this region. This indicates that changing the conditions for EEIs above 50 hPa does not affect the climatological wave forcing from lower levels, and it will enable us to answer how important the middle and upper stratosphere is for SSW generation compared to the 300-200 hPa (Birner and Albers, 2017) and 100 hPa (Polvani and Waugh, 2004) levels highlighted by previous authors.

    There are substantial wave 1 and wave 2 EP flux divergence components in the stratosphere of both H1 and H2 (panels a

and b in Figs. S2 and S3), which shows that there is a large amount of both wave 1 and wave 2 activity in the two control runs. This results in large numbers of both splits and displacements in both runs (Lindgren et al., 2018). In contrast, removal of EEIs everywhere results in an EP flux divergence completely dominated by the wave number of the forcing, with practically only wave 1 and no wave 2 EP flux convergence in NE1 (Fig. S2 c and d) while the opposite is true for NE2 (Fig. S3 c and d). This result is not surprising: removal of EEIs means that waves can only interact with the mean flow, which excludes the

possibilities of energy transfer between wave numbers. In contrast, there are areas of EP flux divergence of a wave number different from that of the tropospheric forcing just above the transition region in NEt1 and NEt2 (Figs. S2f and S3e), and areas of EP flux convergence in the same region in the wave number of the forcing (Figs. S2e and S3f). This suggests that once EEIs are allowed some of the wave activity is transferred from the wave number of the forcing to the other of the two major stratospheric wave numbers. Another aspect worth noting is that the areas of strongest $F_p$ and EP flux convergence occur

further poleward when EEIs are allowed in the stratosphere compared to when they are not. This does not seem to be a result of changing zonal wind climatology and hence a shift in the structure of the wave guide, since the latitudinal polar night jet shifts between the model configurations are modest (Figs. 2 and S1). Robinson (1988) found that removal of all waves but wave 1 resulted in a more equatorward wave flux compared to his nonlinear model run. He deduced that this difference came from the removal of interactions between wave 1 and shorter waves, and that interactions between shorter waves and the mean flow

had a comparably small effect on the wave flux. Our results confirm the conclusion of Robinson (1988) since our experiments allow interactions between all waves and the mean flow, and the equatorward shift persists.

## 4   Impacts on SSWs and polar vortex strength

Table 1 shows the SSW frequencies, time mean and variability of zonal mean zonal wind at 10 hPa and 60° N, and split and displacement ratios for the eight runs. A SSW is defined here as reversal of the zonal mean westerlies at 60° N and 10

hPa (Charlton and Polvani, 2007). The SSW frequency is one way to characterize the variability of the stratospheric polar vortex, but the time mean and variability of the zonal wind at the same pressure level and latitude provide further metrics for how changes in EEIs affect the polar vortex strength. The wave amplitude classification (WAC) introduced by Lindgren et al. (2018) was used to classify the SSWs as splits or displacements.

    Removal of EEIs affects SSW frequencies significantly when the model is forced by wave 1 heating. The SSW frequency in

H1 is 0.66 SSWs per 100 days, but in NE1 the frequency is increased to 0.82 (a 24 % increase). The frequencies are lower in the mixed runs: 0.44 in NEt1 and 0.31 in NEs1 (decreases of 34 % and 53 % compared to the control run, respectively). It is reasonable that the SSW frequency in the wave 1 runs should be affected by removal of EEIs since EEIs affect the climatological





wave forcing quite strongly with this heating perturbation. Figure S4 shows that there are strong tropospheric wave 1 and wave 2 components of $F_p$ in H1, but that the wave 2 component disappears when EEIs are not allowed in the troposphere. However,

the results from the mixed runs are more surprising: removal of EEIs in the middle and upper stratosphere only (NEs1) decreases the SSW frequency by over 50 % compared to the control run. Similarly, allowing EEIs in the middle and upper stratosphere only (NEt1) decreases the SSW frequency by 45 % compared to NE1. As was mentioned in the previous section, the tropospheric and lower stratospheric wave forcing depends on whether or not EEIs are allowed at these levels, and not on the conditions in the middle and upper stratosphere. The fact that the wave forcings and climatologies below 50 hPa in

H1 and NEs1 as well as NE1 and NEt1 are practically identical while their SSW frequencies are very different shows that EEIs in the stratosphere play a major role in SSW generation. Supporting the conclusions of Hitchcock and Haynes (2016), it also highlights the fact that the upper stratosphere is not a passive recipient of wave forcing from below, even though the importance of tropospheric and lower stratospheric wave forcing for SSW generation has often been emphasized (Polvani and Waugh, 2004; Birner and Albers, 2017). Like O'Neill and Pope (1988), we find that SSWs cannot simply be considered forced

by wave-mean flow interactions from a lower boundary. However, there is no clear answer to how EEIs in the middle and upper stratosphere influence SSW frequencies with wave 1 forcing: a comparison between H1 and NEs1 suggests that EEIs are necessary in the middle and upper stratosphere to get high SSW frequencies, while the results for NE1 and NEt1 indicate that allowing EEIs in the middle and upper stratosphere decreases the SSW frequency.

In contrast to the runs with wave 1 forcing, removal of EEIs in the troposphere and lower stratosphere does not dramatically

alter the SSW frequency when wave 2 forcing is used: NE2 and NEt2 have SSW frequencies of 0.51 and 0.45, compared to 0.48 in the control run. This is not surprising considering the fact that almost all tropospheric and lower stratospheric wave forcing in H2 is in the wave number of the forcing (Fig. S5), so removal of EEIs does not affect the climatological forcing as strongly as when wave 1 forcing is used. However, when EEIs are removed in the middle and upper stratosphere only (NEs2) the SSW frequency increases by 37 % compared to H2. This increase in SSW frequency could be the reason for the weakened

climatological polar night jet seen in Fig. 2c, although another explanation is that removal of EEIs weakens the polar night jet. This weakened jet would then require less wave forcing to create SSWs, which would increase the SSW frequency. Like the case of the mixed runs with wave 1 forcing, the increase in SSW frequency shows that middle and upper stratospheric EEIs play a major role in SSW generation. Interestingly, this change in SSW frequency is completely different from the one between NE1 and NEs1: with wave 2 forcing the SSW frequency increases without EEIs in the middle and upper stratosphere, while

the opposite is true with wave 1 forcing.

Unlike the SSW frequencies, some changes in mean polar vortex strength and variability, as measured by the standard deviation of polar vortex strength, are consistent for model runs with both wave 1 and wave 2 forcing. In both cases the standard deviation of polar vortex strength is highest in the control run (H1 and H2), and lowest when EEIs are removed in the troposphere and lower stratosphere (NEt1 and NEt2). As can be expected, low mean polar vortex strengths and high standard

deviations are correlated with high SSW frequencies. More revealing than these numbers are the time evolutions of polar vortex strength, seen in Fig. 3 for all eight model runs during 2000 days of the model simulations. The 2000 days are typical for the model runs. The data has been smoothed with a 10 day filter for clarity, and Fig. S6 in the supporting information shows the





same data unfiltered. Figure 3a shows how removal of EEIs affect the time evolution of the polar vortex strength when wave 1 heating is used. The strength of the polar vortex is highly variable in H1 (black), and has maximum and minimum strengths

higher and lower than any of the other three model runs. Table 1 indicated that the polar vortex strength in NE1 (red) is both lower and less variable than H1, and Fig. 3a confirms this. The variations also seem to be of shorter duration in NE1 compared to H1. Based on the figure, the polar vortex strength in NEs1 (blue) is much more similar to that in H1 than to the one in NE1. The polar vortex in NEs1 exhibits the same long term variability as H1, but does not become strongly negative as frequently as the one in H1. In Fig. 3a the polar vortex in NEt1 (green) exhibits little variability. From the figure it may seem like SSWs

are infrequent in NEt1 compared to the other model runs, but it actually has a higher SSW frequency than NEs1. Instead, the polar vortex NEt1 exhibits a lot of short term variability that is filtered out in Fig. 3a, but is visible in Fig. S6a.

The latitude of maximum polar vortex strength changes slightly between the model runs, and the relative mean values and variabilities of polar vortex strength therefore change when the latitude of interest is changed. However, the timescales of variability are qualitatively similar for given model runs with modest changes in latitude and pressure. Figures 3 and S6 can

therefore be thought of as reasonable representations of overall polar vortex behavior.

As in the case of H1, the polar vortex strength in H2 has a larger variability than in any of the model runs with EEIs removed (Fig. 3b). The polar vortex strength in H2 varies on timescales up to 1000 days; much longer than any other model runs. As in the case of NE1 and H1, the polar vortex in NE2 has a lower mean strength and lower variability than that in H2, and the timescale of the variability is much shorter than in H2. Unlike the case with wave 1 heating, the polar vortex in NEs2 has a

mean strength and variability that resembles that of NE2 more than that of H2. Just like in the case of NEt1, the variability of polar vortex strength in NEt2 is low, and much of the variability happens on short timescales which are filtered away in Fig. 3. However, unlike the case of NEt1 the polar vortex strength in NEt2 does not seem to vary on long timescales.

Figure 3 tells us much about how EEIs affect polar vortex strength and, by extension, SSW frequencies. The structure of the polar vortex variability in NEs1 is similar to that of H1, indicating that EEIs in the troposphere and lower stratosphere are

important for much of the long term (a few hundred days) variability of polar vortex strength when wave 1 heating is used. The fact that the SSW frequency is much lower in NEs1 compared to H1 could indicate that middle and upper stratospheric EEIs are important in order to strongly disturb the polar vortex, at least when the lower stratospheric wave forcing is obtained with wave 1 heating in the presence of EEIs. The similarities in polar vortex behavior between NE2 and NEs2, on the other hand, indicate that EEIs in the troposphere and lower stratosphere are not as important when it comes to variability of the polar vortex

(although the SSW frequency is higher in NEs2 compared to NE2). This is likely due to the fact that most of the tropospheric wave forcing in all model runs with wave 2 heating is in wave 2 (Fig. S5). This is not the case with wave 1 heating, where the tropospheric wave forcing has substantial wave 1 and wave 2 components in H1 and NEs1 but not NE1 and NEt1 (Fig. S4). The differences between H2 and NEs2 show that EEIs in the middle and upper stratosphere are crucial in setting the long term behavior of the polar vortex when wave 2 heating is used. EEIs seem to strongly alter the structure of the stratosphere, which

may alter the amount of wave forcing that can propagate up from below. The fact that the polar vortices in NEt1 and NEt2 exhibit the lowest amounts of variability, and variability on the shortest timescales, could indicate that much of any anomalous tropospheric and lower stratospheric wave flux is damped away in the transition region between 50 and 30 hPa. Comparisons





of total EP flux convergence (not shown) support this: compared to NE1 and NE2, there is stronger EP flux convergence in the transition region of NEt1 and NEt2, and less EP flux convergence further up in the stratosphere. The wave convergence in

the transition region could produce low frequency variability in the polar vortex strength and leave less wave flux to converge further up in the stratosphere, where it would likely affect the polar vortex more strongly.

Table 1 also contains the numbers and fractions of splits in the model runs. 59 % of SSWs in H1 are splits, and this number is increased to 80 % in NEt1. This result seems counterintuitive: as was mentioned above there is strong climatological tropospheric wave 1 and wave 2 flux in H1, while the tropospheric forcing is almost pure wave 1 when EEIs are not allowed.

A possible explanation for this is that much of the wave activity is transferred from wave 1 to wave 2 in the upper stratosphere, where EEIs are allowed. Panels e and f in Fig. S2 show the wave 1 and wave 2 EP flux divergence for NEt1. The panels show that while the EP flux convergence in the upper stratosphere is certainly dominated by the wave 1 component (panel e), much of this wave 1 convergence is overlapped by large wave 2 divergence (panel f). These areas, which can be found above the transition region, likely show regions of wave 1 to wave 2 energy transfer. This energy transfer from wave 1 to wave 2

could supply enough wave 2 forcing to produce splits. The wave 2 vertical flux and flux convergence is lower than its wave 1 counterparts, which shows that the split and displacement ratios are not simply results of the relative climatological forcings. Another explanation for the wave 2 structures is that they arise through barotropic instability, which has been shown to induce wave 2 growth in the stratosphere. Hartmann (1983) investigated the barotropic instability of the polar night jet and suggested that the presence of wave 1 forcing may enhance the growth rates of shorter waves of similar phase speeds. Motivated by

observations of wave 2 growth confined to the Southern Hemisphere winter stratosphere, Manney et al. (1991) investigated the possibility of barotropic instability as a mechanism for wave 2 growth using a barotropic model as well as a zonally symmetric three-dimensional model. They found that both wave 2 and 3, but in particular wave 2, were destabilized when the basic flow in the barotropic model contained stationary wave 1 and zonal flow. Unstable modes of wave 1, 2 and 3 were found in the three-dimensional model, and the authors noted that the wave 2 modes were usually the most unstable.

The fractions of splits for NE1, NEs1, NE2 and NEs2 are in brackets to signify that the SSWs in these runs do not look like typical splits and displacements, even though our SSW classification sorts these SSWs into one of the two categories. Figure 4 shows the absolute vorticity at 10 hPa and 80 hPa on the central dates of typical SSWs in the eight model runs. Corresponding videos showing 60 days surrounding the SSWs can be found in the supporting information. The 10 hPa level is of interest since that is where SSWs are usually defined, and the 80 hPa level was chosen because it is below the transition

region. Panels a and e show a displacement in H1. The displacement of the polar vortex from the pole can clearly be seen at 10 hPa, while there is little to no suggestion of a displacement at 80 hPa. A clear split in H2 can be seen in panels i and m, and the split of the vortex extends all the way down to the lower stratosphere. In contrast, the 10 hPa levels in NE1, NEs1, NE2 and NEs2 (panels b, d, j and l) do not show either splits or displacements, even though wave 1 and wave 2 zonal structures can be seen. Instead it seems SSWs are followed by meridionally oriented waves when the effects of zonal EEIs are removed.

These structures arise when the meridional shear of the flow $(u(\phi))$ interacts with zonal wave numbers. Even though these are wave-wave interactions they are allowed to occur, since our method only removes interactions between zonal waves. Shepherd



(1987) argued that interactions between stationary and transient waves in the observed atmosphere could be understood to first order through processes like these, where meridional shear transfers enstrophy along lines of constant zonal wave numbers.

Even though typical splits and displacements do not occur when EEIs are not allowed in the middle and upper stratosphere,
SSWs in NE1, NEs1, NE2 and NEs2 are still classified as "splits" and "displacements" since the SSW classification of Lindgren et al. (2018) is based on wave amplitudes of geopotential height. The numbers in brackets in Table 1 therefore show that wave 2 amplitudes completely dominate SSWs when wave 2 forcing is used, with NE2 and NEs2 producing 100 % and 90 % wave 2 dominated SSWs ("splits"). These numbers can be explained by the fact that most of the stratospheric wave flux is of wave 2: the upper stratospheric wave 2 flux in NE2 is an order of magnitude stronger than its wave 1 counterpart (Figs. S5c and d),
and although the differences are not as large in NEs2 wave 2 forcing still dominates (Figs. S5g and h). Just like H1 and NEt1, SSWs with wave 1 forcing are also mostly wave 2 dominated, with 57 % and 80 % "splits" for NE1 and NEs1. As in the case of NEt1, this cannot be explained by the dominant wave number of the stratospheric wave flux since wave 1 is the dominant wave number (Figs. S4c, d, g and h). That NE1 produces mostly wave 2 dominated events even though the tropospheric forcing is wave 1 and EEIs are not allowed suggests that enhanced growth of wave 2 in the presence of wave 1 forcing through barotropic
instability, as hypothesized by Hartmann (1983) and shown by Manney et al. (1991), could be a major factor in the wave 2 structures seen in the wave 1 runs.

Figure 4 also shows that EEIs are only needed locally for typical splits and displacements to form. First, true splits and displacements do not occur in NEs1 or NEs2 even though the 80 hPa structures look similar to those of the control runs (panel e and m, compared to h and p). As has already been established, the climatological tropospheric and lower stratospheric wave
forcing is very similar between these two runs. Second, NEt1 and NEt2 show that splits and displacements do occur when EEIs are removed in the troposphere and lower stratosphere, just as long as EEIs are allowed above. The structures of the 80 hPa levels in NE1 and NEt1 (panels f and g) as well as NE2 and NEt2 (panels n and o) are very similar while the 10 hPa levels (panels b versus c, and j versus k) are completely different, with NEt1 producing a displacement and NEt2 a split.

The fraction of splits in H2 was 74 %. When the effects of EEIs are removed in the troposphere and lower stratosphere
(NEt2) the model only produces splits. While almost all climatological tropospheric $F_p$ in H2 is in wave 2, the small amount of wave 1 that does exist (Fig. S5a) is apparently enough to make about every fourth SSW a displacement. Without EEIs in the troposphere and lower stratosphere this low level wave 1 forcing disappears (Fig. S5e). There is almost no climatological wave 1 EP flux convergence in NEt2 (Fig. S3e), suggesting that there is very little wave 1-mean flow interaction.

## 5 Discussion and conclusions

In this paper we have investigated the effects of EEIs on SSW formation in an idealized GCM, and found that removal of EEIs can change the SSW frequency dramatically. While SSWs can be considered wave-mean flow interactions to first order, our results show that removing or adding EEIs alter the conditions for SSW generation in non-predictable ways. While removing EEIs everywhere and below 50 hPa with wave 2 forcing (NE2 and NEt2) does not change the SSW frequency drastically, removal of EEIs in the upper stratosphere only (NEs2) increases the SSW frequency by 37 % compared to the control run.





Since we showed that the wave forcing and climatology of the troposphere and lower stratosphere was dependent on whether or not EEIs were allowed in that same region, this 37 % increase can be attributed entirely to changes in nonlinear interactions in the upper stratosphere. The SSW frequencies with wave 1 forcing are strongly dependent on EEIs: even though H1 and NEs1 as well as NE1 and NEt1 have almost identical tropospheric and lower stratospheric wave forcings their SSW frequencies are very different, with a 53 % decrease in NEs1 compared to H1 and a 45 % decrease in NEt1 compared to NE1. The results

from these mixed runs can be contrasted to previous work which has emphasized the importance of tropospheric and lower stratospheric wave flux for SSW generation (Polvani and Waugh, 2004; Birner and Albers, 2017). The results in this paper confirms previously published results showing that the upper stratosphere is not a passive recipient of tropospheric and lower stratospheric wave forcing (Hitchcock and Haynes, 2016), and that stratospheric nonlinear processes are important for SSW generation (O'Neill and Pope, 1988).

While previous authors have investigated the effects of nonlinear interactions in stratospheric dynamics and found them to be important, this work is the first that has explicitly investigated the effects of EEIs on SSW frequencies in a global primitive equation model. We find that removal of EEIs does not simply increase or decrease the SSW frequency: even with the same tropospheric forcing removal of EEIs in the upper stratosphere can decrease SSW frequencies (H1 compared to NEs1) or increase it (NE1 compared to NEt1). To better understand the effects of EEIs on the polar vortex we also investigated

the variability of polar vortex strength. We find that EEIs in the troposphere and lower stratosphere determine much of the long term stratospheric polar vortex variability when wave 1 heating is used. In contrast, EEIs in the troposphere and lower stratosphere have a small impact on polar vortex variability when wave 2 heating is used, while middle and upper stratospheric EEIs are crucial in determining the long term variability of polar vortex strength. Furthermore, the model runs where EEIs are removed in the troposphere and lower stratosphere but allowed above (NEt1 and NEt2) produce high frequency variability

in polar vortex strength, but little long term variability. It seems that much of the wave forcing from below converges in the transition region between no EEIs and EEIs allowed, which could be the source of the high frequency variability. The frequent dissipation of wave forcing at lower pressure levels could result in less wave forcing in the middle and upper stratosphere, causing lower amounts of low frequency variability in polar vortex strength.

    Some changes caused by removing EEIs can be found in all model runs: the stratospheric vertical wave flux and wave flux

convergence is further equatorward when EEIs are not allowed in the middle and upper stratosphere. This is not a result of a shift in the stratospheric polar vortex, since latitudinal changes in the polar night jet locations are small compared to the changes in wave flux. The equatorward shift is consistent with the results of Robinson (1988).

    We showed that the polar vortex creates meridionally oriented waves instead of splitting or being displaced when our method of removing EEIs is used. We also found that splits and displacements occur even when EEIs are not allowed in the troposphere

and lower stratosphere, indicating that only middle and upper stratospheric EEIs are necessary for split or displacement formation. These results are in contrast to those of Lordi et al. (1980), who found realistic wave 1 patterns in polar stereographic geopotential height when they used wave 1 forcing without EEIs. This discrepancy comes from the difference in methods used to remove the effects of EEIs: many authors, including Lordi et al. (1980), used zonal truncation to only allow one wave number to interact with the mean flow, while we removed all zonal EEIs. The meridional waves arise from interactions between the

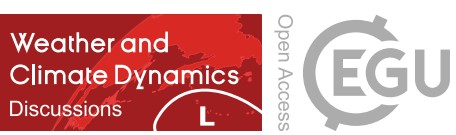

meridional shear of the flow and zonal wave numbers. Lordi et al. (1980) therefore found that wave 1-mean flow interactions are enough to create a displacement in the polar region, while we find that removing EEIs result in SSWs that are neither splits nor displacements.

    With wave 2 forcing, all SSWs were splits when EEIs were turned off in the troposphere and lower stratosphere (NEt2). This is likely due to the fact that all tropospheric forcing is in wave 2 when EEIs are turned off in the lower levels, and the
resulting SSWs are therefore splits. Even though splits and displacements do not occur without EEIs in the middle and upper stratosphere the SSWs in these model runs are still dominated by zonal wave 1 or wave 2 geopotential height anomalies. SSWs are strongly dominated by wave 2 anomalies in NE2 and NEs2, with 100 % and 90 % wave 2 dominated SSWs ("splits"). As in the case of NEt2, this can be explained by the fact that the stratospheric wave flux is mostly of wave 2 format in these model runs. The fraction of splits increased when EEIs were turned off in the lower levels with wave 1 forcing: 80 % splits
in NEt1 compared to 59 % in H1. This is despite the fact that there is strong climatological tropospheric wave 1 and wave 2 wave flux in H1, while the flux is almost purely wave 1 in NEt1. The wave 2 forcing required to produce these splits could originate in the transition region between no-EEIs and EEIs-allowed, where some of the wave 1 flux is transferred to wave 2. Barotropic instability is also likely a factor: Hartmann (1983) suggested and Manney et al. (1991) demonstrated that waves of zonal wave number 1 may enhance growth rates of shorter waves. The strong wave 1 forcing in the lower stratosphere of NEt1
could make the flow unstable to wave 2, which would contribute to the large number of splits. SSWs in NE1 and NEs1 are also mostly wave 2 dominated with 59 % and 80 % "splits" even though the stratospheric wave flux is mostly wave 1. The fact that NE1 produces wave 2 dominated events even though the forcing is of wave 1 and EEIs are not allowed strongly suggests that barotropic instability could be responsible for the wave 2 structure around SSWs in this model run.

*Code availability.*   The Flexible Modeling System (FMS) that is used in this paper can be found at https://www.gfdl.noaa.gov/fms/.

*Data availability.*   Zonal mean zonal winds, geopotential height wave amplitudes as well as 10 and 80 hPa absolute vorticity fields are archived at https://doi.org/10.17605/OSF.IO/PR63B.

*Video supplement.*   Movies of the SSWs seen in Fig. 4 are archived at https://doi.org/10.17605/OSF.IO/3VRME.

*Author contributions.*   Erik Lindgren performed the simulations and data analysis, and wrote the first draft of the manuscript. Aditi Sheshadri originated the idea for the project, contributed to the interpretation of the results, and improved the final manuscript.



*Competing interests.* The authors declare that they have no conflict of interest.

*Acknowledgements.* We thank Paul O'Gorman, Edwin Gerber and Alan Plumb for useful discussions. This work was partially supported by the MIT ODGE Donald O'Brien 2016-2017 Fellowship to Erik Lindgren, NASA Grant NNX13AF80G to Alan Plumb, and the Junior Fellow award 354584 from the Simons Foundation to Aditi Sheshadri.





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





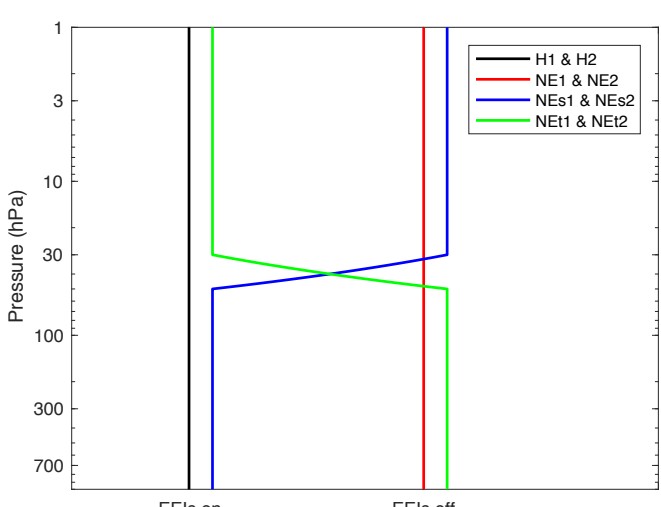

**Figure 1.** Vertical structure of the four runs used for each wave number (1 or 2). See text for details.



**Figure 2.** Zonal mean zonal winds for H2 (a), NE2 (b), NEs2 (c) and NEt2 (d), with panels showing the pressure levels where EEIs are allowed. The contour interval is 5 ms$^{-1}$.





**Figure 3.** Zonal mean zonal wind at 10 hPa and 60° N for H1/H2 (black), NE1/NE2 (red), NEs1/NEs2 (blue) and NEt1/NEt2 (green) for model runs with wave 1 (a) and wave 2 (b) heating. The data has been smoothed with a 10 day filter. The magenta line marks $0\,\mathrm{ms^{-1}}$.





**Figure 4.** Absolute vorticity at 10 hPa and 80 hPa on the central dates of an SSW with wave 1 heating (top two rows) and wave 2 heating (bottom two rows). Displacements can be seen in (a) and (c), while (i) and (k) show splits.



**Table 1.** SSW frequencies, time mean and variability of $\overline{u}$ at 10 hPa and 60° N, and classifications for the model runs.

| Model run | H1 | NE1 | NEt1 | NEs1 |
|---|---|---|---|---|
| Total SSWs (SSWs per 100 days) | 199 (0.66) | 247 (0.82) | 132 (0.44) | 93 (0.31) |
| $\overline{u}_{1060}$, mean (standard deviation) [ms$^{-1}$] | 23 (16) | 17 (12) | 21 (10) | 30 (14) |
| Total splits (fraction) | 118 (0.59) | [143 (0.57)] | 105 (0.80) | [74 (0.80)] |
| Model run | H2 | NE2 | NEt2 | NEs2 |
| Total SSWs (SSWs per 100 days) | 145 (0.48) | 153 (0.51) | 134 (0.45) | 199 (0.66) |
| $\overline{u}_{1060}$, mean (standard deviation) [ms$^{-1}$] | 39 (27) | 33 (19) | 32 (16) | 26 (19) |
| Total splits (fraction) | 108 (0.74) | [153 (1)] | 134 (1) | [179 (0.90)] |