# Peer review of "The Role of Wave-Wave Interactions in Sudden Stratospheric Warming Formation"

_Weather and Climate Dynamics, 2019_

## Referee Comment (RC1) · Anonymous Referee #1 · 21 Oct 2019

Using an idealized model, the paper assesses the role of eddy-eddy interactions (EEIs) on the sudden stratospheric warming (SSW) frequency and the number of splits and displacements. The paper shows that EEIs in the middle and upper stratosphere are important for SSW frequency, but whether the EEIs act to increase or decrease the frequency depends on the wavenumber of the forcing. Moreover, the paper shows that EEIs are important in the troposphere and lower stratosphere for displacements under wave-2 forcing, but that splits and displacements can form under wave-1 forcing without EEIs in the troposphere and lower stratosphere.

I found the paper to be well written and interesting to read and therefore think it will be

worth publishing after the authors have addressed my specific comments below.

Specific comments:

1.) P2, L52-55 and more generally: Do the authors understand why forcing by heating perturbation vs. forcing by topography produces different SSW response with regards to split and displacement ratios? It looks to me that the applied heating perturbation (fig1 in Lindgren et al., 2018) is partially at the tropopause or even in the lower stratosphere in the polar regions. Could it be that applying the heating perturbation at the tropopause, where there is a sharp jump in the stratification, leads to a more non-linear behavior (and hence a larger role for eddy-eddy interaction) than a topographic forcing applied at the lower boundary? Do the authors expect the findings in their paper to be the same if forced by topography instead (where the eddy-eddy interactions might not be as important)? Some discussion on this, and why the use of heating perturbation instead of topographic forcing might be more realistic, would be insightful.

2.) It would be helpful if the authors would state more clearly what they mean by eddy-eddy interactions. In the real stratosphere there are also shorter scale gravity waves, that are not resolved in the model at T42 truncation, whose interaction with planetary waves might be important. The manuscript seems to only describe wave-1 interaction with wave-2. The authors should therefore stress that by EEIs they mean wave 1 and 2 EEIs. Additionally, is there any evidence to suggest that EEIs with wave-3 are important (Figure 4h suggests that there is a wave-3 signature)? The authors could plot EP flux and $F_p$ for other wave components and compare magnitudes to those in Figs. S2-S5.

3.) The authors suggest in several places throughout the manuscript that barotropic instability may explain wave-2 structures in simulations forced by wave-1. The authors could easily verify if the necessary condition for barotropic/baroclinic instability is satisfied in their simulations to back up the statement and make the paper stronger.

4.) Would the results regarding SSW frequency and the role of eddy-eddy interactions be the same if instead of reversal of zonal wind at 60N and 10hPa, a sudden

stratospheric deceleration events based on 2-sigma threshold (as in Birner & Albers, SOLA, 2017 and de la Camara et al, JClim, 2019) were used as a criteria? As the authors discuss (figs. 1 and 3) the climatology and variability of the polar vortex changes quite a bit as a result of removing/allowing eddy-eddy interactions in the stratosphere/troposphere. Therefore a deceleration event criteria might be less subjective.

More technical comments:

P2, L30: "climate" -> "climate variability" as SSWs do not necessarily affect the climate itself, as is shown in your Fig. 2 (i.e., while the frequency of SSWs change as a result of allowing/removing EEIs, the surface climate does not).

P2, L43-44. Please provide a reference for where a GCM has been extensively tuned to produce realistic SSW frequencies. Unless developed to specifically study SSWs, my understanding is that generally GCMs are tuned to produce realistic climatology. The fact that the SSW frequency is realistic is a by-product of this.

P3, L61: "main" -> "mean"

P3, last paragraph. Model resolution is surely another limiting factor of previous work that should be acknowledged. Model resolution determines what eddy-eddy interactions can be captured (see also my main point 2).

P4, L103: remove parenthesis around "Lindgren et al., 2018".

P7, L200 and L202: "pressure levels below 50hPa" -> "vertical levels below 50hPa". Otherwise it is confusing since "pressure levels below 50hPa" imply pressures 50hPa and lower, which is not what the authors intend.

P7, L208-209 :"at pressure levels greater than 50hPa" -> "at vertical levels below 50hPa"

Since Figs. S2-S5 are referenced quite a lot in the main text, it would make sense

to put them in the main body rather than supplementary information. Also, why is the vertical extent plotted in Figs. S2 & S3 different to that in Figs. S4 and S5?

Table 1: Please explain what do the square brackets around some of the entries mean. While it is explained in the main text, it would be helpful to have this information in the table caption.

P11, L345: "u(\phi)" to me means just zonal wind at latitude \phi and not "meridional shear of the flow". Perhaps better use "du/d\phi"? Or since this is not really referenced elsewhere, I would remove this altogether.

P13, L387: "confirms" -> "confirm".

P13, L402: "lower pressure levels" -> "lower levels".
* * *

---

## Referee Comment (RC2) · Anonymous Referee #2 · 21 Oct 2019

Review of 'The role of eddy-eddy interactions in sudden stratospheric warming formation' by Erik Anders Lindgren and Aditi Sheshadri

General Comments: This is an interesting and well-written paper trying to understand the role that nonlinear interactions between waves with different zonal wavenumbers (termed EEIs) can have in the formation of sudden stratospheric warmings (SSWs. Their main conclusions is that middle/upper stratospheric EEIs only have an influence on SSW frequency if the incoming wave flux from the lower troposphere is of a certain type (i.e., wave 1 as opposed to wave 2). They also show that the upper stratosphere is not simply a passive recipient of wave activity from below, but via EEIs can have a key

influence on polar-vortex variability. They further apply their approach to examining the differences in the number of split and displacement type SSWs, finding that when wave-2 forcing is used, EEIs are necessarily required in the troposphere/lower stratosphere to produce displacements, but if wave-1 forcing is used, both splits and displacements are possible without EEIs. My overall suggestion is of minor corrections, which I list below as specific comments and technical comments.

Specific Comments: Line 37-38: Also de la Camara et al. (2019) and White et al. (2019) [both J. Clim] used chemistry-climate GCMs to examine the numbers of SSWs preceded by tropospheric wave activity, finding similar percentages to in Birner and Albers (2017; hereafter BA17). de la Camara et al. (2019) further found the same in the ERA-20C reanalysis.

Line 97: This opening sentence is seemingly not supported by the references provided later in the paragraph. From my understanding, neither BA17 nor Polvani and Kushner (2004) used a model to systematically remove EEIs from the stratosphere. Both used reanalysis datasets. Please clarify.

Lines 155-160: I feel that some reference should be made to Held and Suarez (1994) and Polvani and Kushner (2002) here as this sounds rather like their original setup(s).

Line 161: At which pressure level is the imposed wave-1 or wave-2 heating perturbation cutoff? Please include here as it likely has an influence on the removal of EEIs in the lower stratosphere. From my understanding of Lindgren et al. (2018), the heating perturbation reached up to ∼200hPa; is that correct here too? Further to this, I wonder why the authors did not force planetary waves using topography. Using heating which extends up to the lower stratosphere (at high latitudes), seems like it could have an influence on the removal of the EEIs, especially given the jump in stratification across the tropopause which plays a key role in monitoring the amount of wave activity which can propagate into the stratosphere (Chen and Robinson 1992 – the tropopause valve idea).

Line 194: A citation of Birner et al. (2013; GRL) may be appropriate here who showed the importance of enstrophy fluxes (eddy-eddy interactions) in the region of the subtropical-midlatitude jet.

Lines 204-231: This is a lot of discussion for the supplementary figures. I would consider including some, if not all, of these EP-flux figures in the main article itself rather than the reader continually flicking between the main and supplementary texts.

Lines 323-324: This apparent transfer between waves 1 and wave 2 could be quantified using the enstrophy flux term in the potential enstrophy budget. Indeed, Smith (1983) examined this budget for all low wavenumber waves during a particular SSW event. It even seems that it would be possible using your temperature tendency equation (just by decomposing the final term in the right hand side of eq 1 into different wavenumbers).

Lines 349-351: This seems to suggest therefore that using an algorithm based on the absolute vorticity or potential vorticity may be more apt here. The algorithm used cannot capture proper splits or displacements, which I would think would still occur despite the lack of EEIs. On lines 362-363, you state that 'true splits and displacements do not occur in NEs1 or NEs2' – is this really the case? Or is it actually the case that your algorithm is not picking them up?

Lines 359-361: This is the second time that barotropic instability has been mentioned as a possible candidate for the results found (as well as further mentions in the discussion section). I suggest for the authors to calculate the zonal-mean meridional PV gradient which should not be that difficult as you already have the absolute vorticity shown in Figure 4. From this, one could deduce pretty quickly whether there is any instability using the Charney-Stern criterion (Charney and Stern 1962).

Technical Comments: Line 18: add 'a few' before 'days'

Line 19: add 'can' before 'migrate'

Line 55: 'forced in the troposphere'

Line 61: 'main' > 'mean'

Line 112: 'in an idealized'

Line 131: Please clarify what is meant by 'SSWs, splits and displacements'. Currently it doesn't read well.

Figure 2 and all other figures: please make the label axis, contour labels and colorbar labels etc bigger. They are difficult to see currently.

Line 234: Is this definition also using the extra criterion suggested by CP07? Namely, that SSWs are spaced sufficiently far apart to assume independence (they used 20-30 days I think)? The further conditions (regarding whether winds reversed back to westerly before April 30th, I presume, are irrelevant here).

Line 239: Was the difference in SSW numbers tested using a significance test? Line 283-284: i.e., there are less SSWs, as shown in Table 1.

Lines 287-288: Can you quantify this a little more? Over which latitude range does the jet move between runs? Do the SSW frequencies remain approx the same (perhaps that is what you meant by 'variability of polar vortex strength')?

---

## Author Comment (AC1) · 4 Dec 2019

The comment was uploaded in the form of a supplement:
http://www.weather-clim-dynam-discuss.net/wcd-2019-7/wcd-2019-7-AC1-supplement.pdf

---

## Author Comment (AC2) · 4 Dec 2019

The comment was uploaded in the form of a supplement:
http://www.weather-clim-dynam-discuss.net/wcd-2019-7/wcd-2019-7-AC2-supplement.pdf

---

## Author Response (AR2)

Co-Editor Decision: Publish subject to minor revisions (review by editor) (23 Jan 2020)
by Thomas Birner
Comments to the Author:
Dear Drs. Lindgren, Sheshadri, your answers to the (mostly minor) reviewer comments look fine, but I did want to follow up on a few points and also wanted to bring up some other points that I noticed. Please make appropriate revisions, after which this paper should be acceptable for publication in WCD.

We thank Dr. Birner for taking the time to review the paper. The paper has been edited based on the suggestions of the editor. Line numbers refer to lines in the tracked-changes version of the manuscript.

1. heating perturbation reaching 200 hPa: I don't find the figures comparing H2 and T2 in the response particularly convincing as they show the vertical EP flux in linear scale, which emphasises the troposphere (btw, is this total EP flux or just wave 2? the latter would be the appropriate quantity I think). The structure of vertical EP flux around the tropopause is almost impossible to tell from this scaling and because the strength of EP flux there is at the lowest contour level it is difficult to tell how big the difference between the two runs really is there (from a relative measure perspective the difference could be order 1). It does seem to me, as the reviewers both suggested, that with the setup of the heating perturbations you are putting some of the wave forcing into the (lowermost) stratosphere (this part therefore does not have to propagate through the tropopause). There is also evidence for this in Fig. 6b, where one can see a local minimum of wave 2 upward EP flux near 300 hPa (60 N). It'd be good to mention and discuss the implications of this setup a bit more.

The figure below shows $F_p$ for H2 and T2 between 340 hPa and 35 hPa. The contour intervals for H2 and T2 are identical (the contour interval for the bottom row is greater than for the top row). The top row shows the full $F_p$ for H2 and T2 without any scaling, while the bottom row only shows the wave 2 component with a $p_0/p$ scaling, where p0 is surface pressure (the same pressure scaling as in Figures 5 and 6). The top row shows the same thing as the figure in the response to referee file, but with a different contour interval and shorter vertical extent.
The top row shows that the magnitude of the flux around the tropopause is comparable between the two runs, although the structures of the fluxes are somewhat different. The bottom row shows only the wave 2 components, and in that case the fluxes look more different. However, the absolute majority of vertical wave flux is in the wave 2 component for T2, while H2 has more flux in wave 1. If one wants to compare the vertical wave fluxes of H2 and T2 we therefore think it is more reasonable to compare the overall fluxes rather than the wave 2 components by themselves.
Nevertheless, we have added some discussion about the implications of the vertical extent of the heating perturbation in Section 2 (L193-198). We believe that the vertical extent of the heating perturbation does not affect our results, since the transition between allowing and not allowing WWIs occurs between 50 and 30 hPa (well above the highest extent of the heating perturbation). This location for the transition region (as opposed to the tropopause) was chosen just because it was well above regions of high wave activity around the tropopause and lower stratosphere. The rather high vertical location of the transition region is why the effects of WWIs are discussed separately in the troposphere and lower stratosphere, and middle and upper stratosphere.

[Figure]

2. meaning of EEIs (reviewer 1, specific comment 2): should you perhaps refer to this as wave-wave interactions (WWIs)? At least to me (and perhaps also to reviewer 1) the term "eddy-eddy interactions" provokes the connotation of smaller scale phenomena such as within a spectrum of scales in turbulence (more appropriate for tropospheric dynamics). In the stratospheric context here the interaction is mainly between two waves as you point out, so WWI may be a bit more to the point.

We have changed "eddy-eddy" to "wave-wave" and "EEIs" to "WWIs" throughout the manuscript, and renamed the model runs accordingly (NE1 to NW1, etc.) The figure labels in the Supporting information document have been edited as well.

More importantly about the removal of EEIs/WWIs: the fact that you only remove these interactions in the zonal direction, and not in the meridional, would be good to emphasise more. Right now this is only mentioned in passing (beginning of section 2) and then briefly near line 370. Otherwise it's surprising why you should get high wavenumber structures in the meridional direction (those barotropically unstable vorticity "ripples"). I do wonder to what extent these high meridional wavenumbers, which occur much less pronounced in the full simulations (and the real atmosphere), may lead to artefacts?

We have emphasized the fact that we are removing zonal WWIs. It is mentioned in the Abstract (L3), Introduction (L125), Section 2 (L156, L158-159, L164-168), Section 4 (L384-385), and the Discussion and conclusions (L461).
We have not investigated the effects of meridional waves on the dynamics in these model runs, but we think it is unlikely that they have a large effect on the overall climatology since they only occur during vortex breakdowns, and not when the vortex is more stable. We have added this statement in the manuscript (L387-389, L464-465).

3. barotropic instability: have you also checked for symmetric/inertial instability? In any case, one question I had on this is that if you find that the necessary condition for barotropic instability is met, why do you not observe barotropically unstable waves develop? Or do you? Perhaps because the non-linear breakdown of the instability requires wave-wave interactions it doesn't end up kicking off? Please discuss somewhere in paper.

Barotropically unstable waves of wave number 2 likely develop in all model runs. We mean that barotropic instability is the most likely reason for the presence of wave 2 in the heating wave 1 model run without wave-wave interactions (NW1). We do not know of any other plausible explanation for the presence of wave 2 in this model run. Barotropically unstable waves of wavenumber 2 develop and grow in the presence of wave 1 forcing, and this is the reason why so many "splits" (wave 2 disturbances) are observed in a model run with pure wave 1 forcing and no wave-wave interactions. Hartmann (1983) and Manney et al. (1991) both showed that barotropically unstable waves of shorter wave numbers (and particularly wave number 2) develop in the presence of wave 1 forcing. This is discussed in the paper (L361-372, L401-404, L476-482).
We have not checked for symmetric/inertial instability.

4. tuning of GCMs to produce realistic SSW frequency: in the case of WACCM the Charlton et al. paper doesn't mention anything about tuning in WACCM, only that its SSW frequency is too low. So unless you have a reference where the tuning in WACCM to produce realistic SSW frequency is explicitly mentioned you should remove this part of the statement and only refer to idealised modelling studies.

The sentence has been edited to refer only to idealized modeling studies (L46-49).

[revised manuscript text omitted]